# Visualization of Three Sclerotiniaceae Species Pathogenic on Onion Reveals Distinct Biology and Infection Strategies

**DOI:** 10.3390/ijms22041865

**Published:** 2021-02-13

**Authors:** Maikel B. F. Steentjes, Sebastian Tonn, Hilde Coolman, Sander Langebeeke, Olga E. Scholten, Jan A. L. van Kan

**Affiliations:** 1Laboratory of Phytopathology, Wageningen University, 6708 PB Wageningen, The Netherlands; maikel.steentjes@wur.nl (M.B.F.S.); s.tonn@uu.nl (S.T.); hilde.coolman@hotmail.com (H.C.); langebeekesander@gmail.com (S.L.); 2Plant Breeding, Wageningen University, 6708 PB Wageningen, The Netherlands; olga.scholten@wur.nl

**Keywords:** infection biology, onion, *Allium cepa*, *Botrytis squamosa*, leaf blight, *Botrytis aclada*, neck rot, *Sclerotium cepivorum*, white rot, fluorescence microscopy

## Abstract

*Botrytis squamosa*, *Botrytis aclada*, and *Sclerotium cepivorum* are three fungal species of the family Sclerotiniaceae that are pathogenic on onion. Despite their close relatedness, these fungi cause very distinct diseases, respectively called leaf blight, neck rot, and white rot, which pose serious threats to onion cultivation. The infection biology of neck rot and white rot in particular is poorly understood. In this study, we used GFP-expressing transformants of all three fungi to visualize the early phases of infection. *B. squamosa* entered onion leaves by growing either through stomata or into anticlinal walls of onion epidermal cells. *B. aclada,* known to cause post-harvest rot and spoilage of onion bulbs, did not penetrate the leaf surface but instead formed superficial colonies which produced new conidia. *S. cepivorum* entered onion roots via infection cushions and appressorium-like structures. In the non-host tomato, *S. cepivorum* also produced appressorium-like structures and infection cushions, but upon prolonged contact with the non-host the infection structures died. With this study, we have gained understanding in the infection biology and strategy of each of these onion pathogens. Moreover, by comparing the infection mechanisms we were able to increase insight into how these closely related fungi can cause such different diseases.

## 1. Introduction

Onion is an important vegetable crop that is cultivated worldwide, but its production is threatened by pathogens and pests. There are three main onion diseases that are caused by species of the fungal family Sclerotiniaceae. *Botrytis squamosa* is the causal agent of onion leaf blight, while *Botrytis aclada*, *Botrytis byssoidea* and *Botrytis allii* cause neck rot, and *Sclerotium cepivorum* causes onion white rot. Although the species are related, the diseases they cause are distinct and have their own etiology.

The most notorious disease is *Botrytis* leaf blight, caused by the necrotroph *B. squamosa*. The fungus was first described nearly 100 years ago, and even today, leaf blight is still a major disease in almost all onion cultivation areas worldwide [1,2,3]. The disease is characterized by small necrotic spots on onion leaves that expand at a later stage leading to blighting of leaves and early leaf senescence, eventually resulting in a reduction of plant growth, bulb yield and quality [4,5]. Compared to neck rot and white rot, the disease cycle of leaf blight is relatively well studied. Spores of *B. squamosa* are dispersed by wind, and upon landing on the onion leaf surface, the spores germinate to penetrate the surface and enter the leaf tissue. Once inside the leaf, the fungus spreads intercellularly and proliferates by obtaining nutrients from killed host cells. Infected onion leaves turn necrotic, and on the outside, new conidia are produced that can initiate a new disease cycle [6,7].

*Botrytis* neck rot is a disease that results in rotting of the neck of the onion bulb. Although infection is initiated in the field, symptoms are observed only after bulbs have been harvested and stored, making neck rot a post-harvest disease [5,8]. Neck rot is caused by a complex of the three species *Botrytis aclada*, *B. byssoidea*, and a hybrid species of the former two, called *B. allii* [9,10]. Since the disease remains asymptomatic during the growing season of onion plants, it is difficult to study its etiology. In the past, symptomless infection of leaves has been hypothesized to lead to an endophytic growth of the fungus toward the bulb [11]. In addition, endophytic growth from contaminated seed has been suggested as an infection strategy, as correlations between the numbers of infected seed and the incidence of neck rot in storage were reported [12,13,14]. Despite these hypotheses, the infection process of neck rot still remains unclear.

White rot of onion is caused by the soil-borne pathogen *Sclerotium cepivorum*, which mainly colonizes roots, basal plates, and bulbs of the onion plant. Symptoms of infected plants are premature yellowing, dieback, wilting of older leaves, and stunting of plants, followed by death of all foliage and sometimes root rot [15,16]. Unlike *Botrytis* species, *S. cepivorum* does not produce spores to spread and establish disease. Instead, spread occurs through sclerotia formed on infected hosts which can remain dormant in soil for up to many years. Root exudates of *Allium* plants have been reported to stimulate sclerotia to germinate and infect the roots [17,18,19,20]. Once a production field is infested with *S. cepivorum*, onion cultivation is no longer possible.

Although all three diseases pose a serious threat to onion cultivation, the infection biology of these fungi, especially the ones causing neck rot and white rot, is still poorly understood. To provide insight into the early infection biology and to understand the interaction between plant and fungus, we developed GFP (green fluorescent protein)-expressing transformants of *B. squamosa*, *B. aclada*, and *S. cepivorum* and visualized the infection process on onion. Using epifluorescence and confocal microscopy, we were able to visualize the interaction between these fungi and onion, and colonization of the host tissues. By comparing infection strategies of the three related fungi, we aimed to determine how they cause such distinct diseases on the same host and aimed to provide insights for the development of new control strategies.

## 2. Results

### 2.1. Transformation of B. squamosa, B. aclada and S. cepivorum with gfp Yields Fluorescently Labeled Fungi

To visualize the infection process of the three different fungi in onion, we used fluorescently labeled transformants. A GFP encoding gene was inserted into the genomes of the fungi using a polyethylene glycol-mediated protoplast transformation, resulting in multiple GFP-expressing transformants of each fungal species. Transformants did not show abnormalities in morphology and pathogenicity and were able to fluoresce (Figure 1a). Furthermore, transformants were not reduced in growth speed as compared to WT (Figure 1b). For each species, one fluorescent transformant with normal morphological appearance and growth rate was selected to study the infection biology.

### 2.2. B. squamosa Enters the Onion Leaf by Growth through Stomata or into Anticlinal Walls

The average width and length of *B. squamosa* conidia measured 15.6 µm and 22.6 µm, respectively, which is in accordance with previously published measurements [21]. Spores of *B. squamosa* applied to the surface of onion leaves germinated within the first two hours after inoculation (Figure 2a). The majority of spores germinated unipolar (Figure 2b), but bipolar germination from opposing sides of the spore was also observed (Figure 2c). Occasionally, germ tubes growing over the onion leaf surface branched, mostly with one but sometimes with two opposing lateral branches (Figure 2d). The width of germ tubes measured on average 5.5 µm. The length depended on the nutrient concentration applied in the inoculum, with higher nutrient concentrations resulting in longer germ tubes at 8 h post inoculation (HPI) on the onion leaf surface (Figure 2e). Spores inoculated without nutrients in the inoculum did not germinate and did not show fluorescence. In addition to length, the curvature of germ tubes and the fluorescence intensity also varied with nutrient concentration in the inoculum (Appendix A).

Most of the germ tubes that penetrated the onion leaf surface did so by entering through stomata (Figure 2f,g). Alternatively, *B. squamosa* was observed to enter the onion leaf by growing into anticlinal cell walls of the onion epidermis (Figure 2h,i). In several cases, germ tubes were observed to grow over stomata, presumably when they were closed, instead of growing through them. Likewise, germ tubes very often grew over anticlinal walls of onion epidermal cells as compared to growth into anticlinal walls toward mesophyll tissue, suggesting leaf penetration by growth into the epidermal layer might be triggered by external stimuli.

### 2.3. Botrytis Aclada Reproduces Asexually without Penetrating the Onion Leaf Surface

*Botrytis* neck rot is a storage disease of onion that does not display symptoms in the field. To resemble field conditions, inoculations of *B. aclada* were performed with spore densities that did not result in disease symptoms or other plant responses. Inoculation of *B. aclada* spores on onion leaves resulted in germination within a similar timeframe and with comparable polarity and branching patterns as for *B. squamosa*. The conidia of *B. aclada* however, measured on average 5.2 µm wide and 7.0 µm long which is similar to previous reports [22,23]. This is over three times smaller in size, and therefore approximately 30 times smaller in volume as compared to *B. squamosa*. In addition, germ tubes were smaller than germ tubes of *B. squamosa*, measuring on average 3.2 µm wide. Germ tubes of *B. aclada* were observed to grow exclusively over the surface of the onion leaf. Even while we examined inoculated leaf samples up to 48 HPI, we neither observed penetration of the onion leaf surface by growth into the epidermal cell layer nor through stomata, even when hyphae grew along or over stomata (Figure 3a,b). Instead, the hyphae branched and developed conidiophores with new conidia from 72 HPI onwards (Figure 3c). No lesion, discoloration, or any other plant response was observed, while the newly developed conidia could be seen by eye (Figure 3d,e). To ensure that the development of conidiophores was not an artifact of nutrients used in the inoculum for synchronized spore germination, spores were pre-incubated in nutrient solution to initiate germination and washed with water before inoculation. Germination, growth, and formation of conidiophores were similar to nonwashed inoculum, indicating that the nutrient for developing conidiophores are acquired from another source than the inoculum. In addition to inoculating *B. aclada* on its host plant onion, we also inoculated nonhost plants. Production of new conidia without necrotic symptoms was observed on leaves of tomato and lily, but not on *Nicotiana benthamiana* (Appendix A).

In order to follow the development of the fungus over a longer time period, onion leaves were inoculated and sampled at weekly intervals and divided into 10 mm segments to examine outgrowth of *B. aclada*. The fungus could never be isolated from any leaf segment other than the segment on which it was inoculated (Appendix A). *B. aclada* could be isolated from the segment with the initial inoculation spot for up to 50 days postinoculation (DPI) even when leaves were already naturally withered of age, suggesting regrowth from newly produced conidia instead of outgrowth of invasive hyphae since the dry and withered leaves unlikely sustain hyphal growth.

### 2.4. Sclerotium cepivorum Infects Onion by Formation of Appressorium-Like Structures and Infection Cushions

To study the infection process of *S. cepivorum,* onion seedlings were inoculated with pregerminated sclerotia of *S. cepivorum*. During germination of sclerotia the hard shell burst, followed by the outgrowth of multiple branched hyphae (Figure 3a). No targeted growth toward or away from the host plant root was observed. At early time points (12 to 24 HPI), *S. cepivorum* hyphae attached to the root epidermis of the onion seedling and developed infection structures. Both appressorium-like structures, formed by swelling of the tip of a single hypha (Figure 4b), and infection cushions (also referred to as compound appressoria), formed by one or two branched and curled hyphae resulting in a bundle of intertwined hyphae (Figure 4c), were observed. Approximately 10 to 12 h after the formation of the infection structures, invasive hyphae were observed growing inside the onion root (Figure 4d,e).

Once inside the root, hyphae grew in cortical tissue while epidermis and hypodermis remained intact (Figure 4f). Hyphae progressing toward the root tip were also observed to grow in or around vascular tissue. In advanced stages, the cortical root tissue developed cavities that were enriched with hyphae. Eventually, the fungus exited the root by growing through the root tip (Figure 4g). 

Inoculation of *S. cepivorum* on the basal part of the leaf resulted in different fungal growth patterns than inoculation on root tissue. After penetration, hyphae grew rapidly throughout the plant. Approximately 24 HPI, hyphae in the epidermal and subepidermal layers had grown from the initial infection site toward all directions (Figure 4h). Host tissue appeared to be macerated around and in front of the leading hyphae (Figure 4i). The fungus further colonized and spread across the width of the basal part of the leaf and grew both toward leaf as well as root tissue (Figure 4j). Approximately 5 DPI the fungus grew into the roots, while all other plant parts were already fully colonized and macerated.

In addition to visualizing the interaction between *S. cepivorum* and its host plant onion, we tested the host specificity of *S. cepivorum* by inoculating seedlings of nonhost plants cabbage and tomato. On cabbage, hyphae grew around plant tissue, but no formation of infection structures was observed (data not shown). On tomato seedlings however, hyphae that emerged from sclerotia attached to the epidermis of roots and root hairs and formation of appressoria and infection cushions were observed within 24 h after inoculation (Figure 5a–c). Both appressoria and infection cushions morphologically resembled the structures observed on onion seedlings. In all cases however, the infection structures lost fluorescence within 10 to 24 h upon contact with the nonhost, indicating dying of the infection structure (Figure 5d–f). Only the apical cells that were in direct physical contact with the tomato-root epidermis or root hairs died, while other hyphae retained their fluorescence and continued growing. In none of the observed cases could hyphae that had penetrated the tomato-root tissue be detected.

## 3. Discussion

In this study, we transformed *B. squamosa*, *B. aclada*, and *S. cepivorum* to obtain fluorescently labeled fungi that allowed for an accurate visualization of the infection processes in onion tissues. This is the first report of transformation of *B. aclada* and *S. cepivorum*, and although *B. squamosa* was the first *Botrytis* species that had been transformed [24], this is also the first report on GFP-expressing *B. squamosa*. Although the *gfp* cassette was ectopically integrated, the successful protoplast transformation paves the way for molecular studies using knockout mutants of these fungi, possibly by adapting the CRISPR-Cas9 protocol developed for *B. cinerea* [25].

When *B. squamosa* was inoculated on onion leaves, two modes of penetration were observed: (1) growth through stomata and (2) growth into anticlinal walls of epidermal cells. This is in accordance with previous observations [6,7,26], but formation of distinct appressoria was not observed by us. *Botrytis* species, including *B. squamosa*, are known to secrete plant cell-wall-degrading enzymes such as cutinases, pectinases, and cellulases that facilitate progressive growth of germ tubes penetrating the leaf surface, thereby diminishing the need to develop distinct appressoria [27,28,29]. When inoculating *B. squamosa* on onion leaves, we observed that the higher the nutrient concentration in the inoculum, the longer the germ tube and the higher the fluorescence intensity, whereas in water, spores did not germinate at all. The availability of nutrients likely influenced the basal metabolism of the fungus, thereby affecting the speed of polarized growth as well as expression of the *gfp* gene. The activity of the *Aspergillus nidulans oli*C promotor that was used to control *gfp* might be influenced by sugar availability, since the *oli*C gene encodes a mitochondrial ATP synthase subunit [30]. 

*Botrytis* neck rot is a postharvest disease of onion that is initiated in the field but only becomes apparent upon storage of the bulbs for several weeks or months, as is typically done for Dutch long-day onions [5,8]. From our leaf infection assays for microscopy, as well as from the long-term infection and resampling assays, we did not observe penetration of the onion leaf surface, nor did we obtain evidence of asymptomatic, endophytic growth of *B. aclada* through onion leaves. This is in contrast to a study performed by Tichelaar et al. [11] who reported germ tubes growing through stomata of onion leaves. In contrast to the latter study, we typically used whole leaves and whole plants in our experimental setup to ensure biologically relevant conditions. Remarkably, we could observe germ tubes growing through stomata after the epidermal layer of the onion leaf was peeled off and placed on agar medium before inoculation. The isolation of the single-cell layer likely resulted in cell death and dysfunctioning of stomata, providing an artificial host substrate that *B. aclada* was able to penetrate.

Necrotrophs like *Botrytis* species are presumed to destruct host cells to obtain nutrients and proliferate. The formation of conidiophores without penetrating the leaf surface and killing host cells suggests that *B. aclada* obtains nutrients from another source. Several fungal plant pathogens, such as certain *Colletotrichum* species and *Rhynchosporium secalis*, have a subcuticular growth phase in which they grow under the cuticle and within the periclinal and anticlinal walls of epidermal cells [31,32]. Our results suggest that *B. aclada* has a similar growth, in which the nutrients required for developing new conidia might be acquired from the cuticle, consisting of a layer of cutin with embedded polysaccharides and an overlying layer of waxes, or the polysaccharides from the outer periclinal cell wall of epidermal cells [33].

There is no consensus about the way in which *B. aclada* ends up in the neck area of the onion bulb. It has been hypothesized that the neck area could be reached through endophytic growth from contaminated seed [14,34], through endophytic or epiphytic growth from asymptomatic infection of leaves [11], or by growth into wounds in the onion bulbs, neck, or leaves [35]. The conidia production that we observed on green leaves of onion and even other plant species may play an important role in the disease cycle. This may especially be the case just before the harvest of onion bulbs, when relatively dry foliage is cut from the bulbs creating a wound just above the neck area and thus providing a potential entry point for conidia of *B. aclada* during the period the bulbs remain on the field [35]. Moreover, the cutting and thus movement of foliage will facilitate spores to become airborne. The simultaneous induction of a temporary high spore pressure and the availability of entry points to the onion bulb might result in the initiation of infection. The capacity of *B. aclada* to also grow and sporulate on the surface of other plants which are considered to be nonhosts (e.g., tomato and lily) might also generate a reservoir of spores that may be present in the vicinity of the crop at the moment when the foliage is cut and the bulb tissue becomes exposed to the air. To what extent nonhost plants serve as a source of *B. aclada* inoculum is a challenging question to address.

To study the infection biology of *S. cepivorum*, onion seedlings were inoculated using pregerminated sclerotia. In the field, sclerotia remain dormant in the soil until a host plant growing nearby secretes root exudates that induce germination [19,20,36,37]. However, several studies have reported that sclerotia can spontaneously germinate without root exudates in sterile conditions, suggesting that root exudates do not directly induce germination but instead might abolish suppression of germination by antagonistic soil microbes [38,39].

To facilitate visualization of the infection process, onion seedlings were grown in liquid and inoculated with germinated sclerotia. Infection progressed rapidly and the infection process observed is in accordance with previous descriptions [40,41,42,43]. Penetration of the epidermis occurred after formation of appressorium-like structures and infection cushions. On roots of cabbage seedlings, no formation of infection structures was observed, possibly because the cuticle composition or the absence of specific root exudates did not induce infection structure formation [44,45,46]. On tomato seedlings *S. cepivorum* formed infection structures similar to those on onion seedlings, although they did not result in successful invasion of the tomato-root system. The abrupt loss of fluorescence of the apical cells of the infection structures indicates that a successful tomato defense response prevented infection, probably because it triggered fungal cell death in infection structures, however without affecting the neighboring hyphal cells of *S. cepivorum*. The cell death in the infection structure suggests a pre-invasive mode of defense, possibly by antifungal secondary metabolites or antifungal proteins. For example, *Arabidopsis thaliana* produces upon infection the phytoalexin camalexin which is able to induce programmed cell death in *B. cinerea* [47]. Likewise, the tomato saponin α-tomatine is known to kill cells of *Fusarium oxysporum* by inducing programmed cell death [48]. Moreover, tomato roots are known to contain chitinases and β-1,3-glucanases with strong inhibitory activity on fungal hyphae, especially on growing tips. However, the observation that infection structures still contained a seemingly intact outer wall devoid of cytoplasm and GFP, would argue against the involvement of hydrolytic enzymes in the successful defense in tomato roots and root hairs against *S. cepivorum*. 

Our research clearly shows that the infection process of onion by *B. squamosa*, *B. aclada*, and *S. cepivorum* involves three completely different infection strategies. Even the sister taxa *B. aclada* and *B. squamosa* showed a completely different biology, despite their close relatedness and their similar genome structure and gene content, with respect to secondary metabolites and effector genes [49]. It is clear that although *B. squamosa*, *B. aclada*, and *S. cepivorum* are related fungal pathogens of onion, they all have their own unique biology and infection strategy and cause distinct diseases. When developing control strategies or breeding for onion cultivars resistant against leaf blight, neck rot, and white rot, it is important to consider the fundamental differences between the causal agents.

## 4. Materials and Methods 

### 4.1. Fungal Isolates and Culture Conditions

*B. squamosa* isolate MUCL31421, *B. aclada* isolate 633, and *S. cepivorum* isolate UFL were used as recipient strains for transformation. Spores of *B. squamosa* were obtained by growth on autoclaved onion leaves on top of water agar (Oxoid, Basingstoke, UK) at a temperature of 20 °C in the dark for 3 days. After 3 days, plates were transferred to 16/8h day/night rhythm of white light supplemented with UV light (330–370 nm, Sylvania F15T8/BLB) at a temperature of 20 °C for 4 days. Spores of *B. aclada* were produced by growth on MEA (Oxoid) in the dark at 20 °C for 7 days, after which plates were exposed to 16/8 h day/night rhythm of white light supplemented with UV light (360–380 nm, Phillips TLD 18W/08) for 3 days. Sclerotia of *S. cepivorum* were obtained by growth on MEA (Oxoid) plates in the dark at 20 °C. For long term storage, spores of *B. squamosa* and *B. aclada* were kept in 15% glycerol at −80 °C, and sclerotia of *S. cepivorum* were stored at room temperature.

### 4.2. Fungal Transformation

To fluorescently label the fungi, a gene cassette containing *gfp* (codon optimized for *B. cinerea*) under control of a *oliC* promotor, together with a hygromycin resistance marker, was amplified from plasmid pNDH-OGG [50]. Using a PEG-mediated protoplast transformation [51], the construct was inserted into the genomes of *B. squamosa*, *B. aclada*, and *S. cepivorum*. Protoplasts of *B. squamosa* and *B. aclada* were obtained by digestion of overnight germinated spores in 1% malt extract (Difco, Leeuwarden, the Netherlands) using Glucanex (Sigma-Aldrich, Zwijndrecht, the Netherlands). For *S. cepivorum*, mycelium was grown on PDA (Oxoid) covered with cellophane for 3 days after which the mycelium was isolated and homogenized (Kinematica CH-6010). The suspension of mycelial fragments was incubated overnight in 1% malt extract (Difco) before protoplasting using Glucanex (Sigma-Aldrich). Transformed protoplasts were selected using an initial hygromycin concentration of 30 µg/mL for *B. squamosa* and *B. aclada* and 75 µg/mL for *S. cepivorum*. Transformant colonies were assessed for their capacity to fluoresce and the integration of the *gfp* gene and resistance cassette was confirmed by PCR. Virulence of transformants was checked by inoculation on onion leaves and bulb scales, and growth speed was assessed by measuring diameter of five replicate colonies per strain on MEA plates and calculating growth rates using trendlines.

### 4.3. Plant Material and Inoculation

All onion plants used in this study were grown in a climate chamber with 12 h light, 70% relative humidity, 18 °C day temperature, and 16 °C night temperature.

For inoculation assays of *B. squamosa*, young but fully grown leaves of 2–4 month onion plants cv. Manesco were used. Detached leaves were placed in humid boxes and the cuticula of leaves was gently wiped with tissue paper to facilitate inoculation. Spores were suspended in 12 g/L PDB (Difco) to a final concentration of 5 × 10^4^ or 10^5^ spores/mL and inoculated in 1 µL droplets.

For inoculation assays of *B. aclada*, leaves of 4–6-weeks-old onion plants from sets of cv. Cupido were used. Spores were suspended in 12 g/L PDB to a final concentration of 10^4^ or 10^5^ spores/mL and inoculated in droplets of 1 µL. For the long-term infection assay, 6-week-old onion sets cv. Cupido were inoculated with one inoculation droplet of 1 µL per leaf with 10^4^ spores/mL in 12 g/L PDB. On weekly intervals, leaves were segmented into 10 mm wide segments and placed on MEA plates with 100 µg/mL hygromycin. Inoculation on nonhost plants species was performed on tomato cv. Motelle, *Nicotiana benthamiana* and a lily Oriental × Asiatic hybrid cultivar.

For inoculation of *S. cepivorum*, seedlings of onion cv. Stuttgarter Riesen, tomato cv. Moneymaker, and white cabbage cv. Express, were placed on object glasses with fixed cover slips with roots and part of the hypocotyl in between the glass slides. The prepared slides were placed upright in 0.25 Murashige & Skoog basal salt mixture (Duchefa, Haarlem, the Netherlands). For inoculation, sclerotia were cleaned and pregerminated on droplets of MEA before they were placed directly onto the seedling. Inoculated seedlings were placed back in nutrient solution and visualized at regular intervals.

### 4.4. Fluorescence Microscopy and Germ Tube Measurements

For determining fluorescence of the transformant colonies, images were captured with a Nikon Eclipse 90i fluorescence microscope mounted with a Nikon cooled camera head DS-5Mc using a Plan Fluor 4×/0.13 lens. For visualization of the *S. cepivorum* infection process, Plan Fluor 10×/0.30 and 40×/0.75 lenses were used. With a Nikon Intensilight mercury lamp, GFP-expressing fungi were visualized using an FGP(R)-BP filter (exitation: 460–500 nm, dichroic mirror: 505, barrier wavelength: 500–550 nm). For visualization of the *B. squamosa* and *B. aclada* infection, a Zeiss LSM 510-META confocal laser scanning microscope equipped with a mmi DCA252cF-K07 CellCamera was used with EC Plan-Neofluor 40×/1.30 Oil DIC and Plan Apochromat 63×/1.4 Oil DIC objectives. An Argon laser 488nm was used for excitation with the GFP signal passing through a BP 505–530 filter and the chloroplast autofluorescence and propidium iodide signal passing through an LP 615 filter. To visualize the epidermal cell walls, inoculated leaf segments were stained in 0.05 mg/L propidium iodide. For the visualization of leaf surface penetration, Z-stacks of optical sections including pathogen, leaf surface, epidermal cells, and mesophyll cells were acquired. In addition, 3D-projects of Z-stacks were made using the image analysis tool imageJ. For the *B. squamosa* germ tube measurements of different nutrient concentrations in the inoculum, in total 24 inoculation spots with four replicates per nutrient concentration were imaged resulting in 448 measurable germ tubes. Using imageJ, a segmented line was manually drawn from the base of the spore until the apex of the germ tube and length, fluorescence (measured as average intensity) and curvature (calculated as the ratio feret value/length) were measured per germ tube. Spore size and germ tube width were measured similarly using 22 images of germinated spores for both *B. squamosa* and *B. aclada*

### 4.5. Statistical Analysis

All statistical analyses were performed in GraphPad 9.0 (Prism, San Diego, CA, USA). WT and GFP-transformants’ growth rates were pairwise compared for *B. squamosa*, *B. aclada*, and *S. cepivorum* using unpaired *t*-tests. Germ tube length, curvature and fluorescence intensity at different concentrations PDB in the inoculum were side-by-side analyzed using unpaired *t*-tests. Differences were considered to be statistically significant with two-tailed *p* values ≤ 0.05.

## Figures and Tables

**Figure 1 ijms-22-01865-f001:**
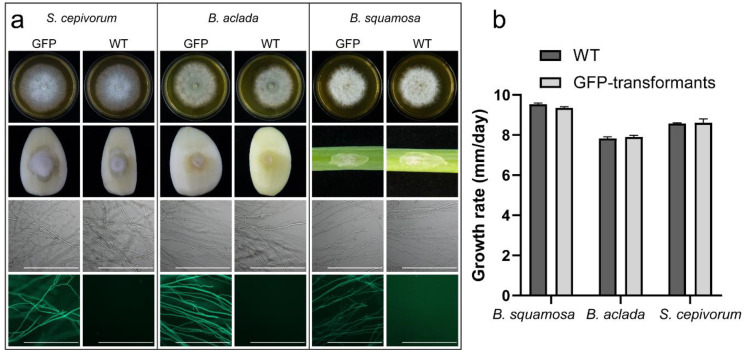
Evaluation of WT and GFP-expressing transformants of *B. squamosa*, *B. aclada* and *S. cepivorum*. (**a**) No differences were observed between WT and transformants based on morphology and growth on agar plates (first row) and pathogenicity on host tissue (second row), and only transformant strains are able to fluoresce (third and fourth row). Scale bar = 500 µm. (**b**) No difference was observed in growth rate between WT and transformants (unpaired *t*-test, *p* > 0.05, error bars representing standard error).

**Figure 2 ijms-22-01865-f002:**
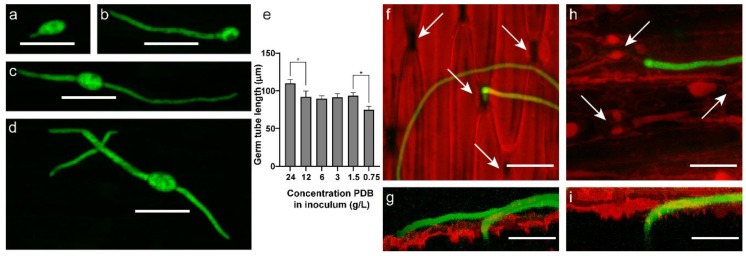
*B. squamosa* growth on and penetration of the onion leaf surface. Fluorescence microscope images of spores germinating (**a**,**b**) unipolar and (**c**) bipolar, and (**d**) a branching germ tube. (**e**) The effect of nutrient concentration in the inoculum on germ tube length at 8 h post inoculation (HPI) on the onion leaf surface, (unpaired *t*-test, * *p* < 0.05, error bars representing standard error). Confocal laser scanning microscopy images of *B. squamosa* penetrating the onion leaf surface. (**f**) Top view and (**g**) front view of a 3D projection of stomatal penetration. (**h**) Top view and (**i**) front view of a 3D projection of growth into anticlinal walls of epidermal cells. All scale bars represent 50 µm, arrows indicate stomata, *B. squamosa*-GFP (green fluorescent protein) in green and propidium iodide-stained onion cell walls in red.

**Figure 3 ijms-22-01865-f003:**
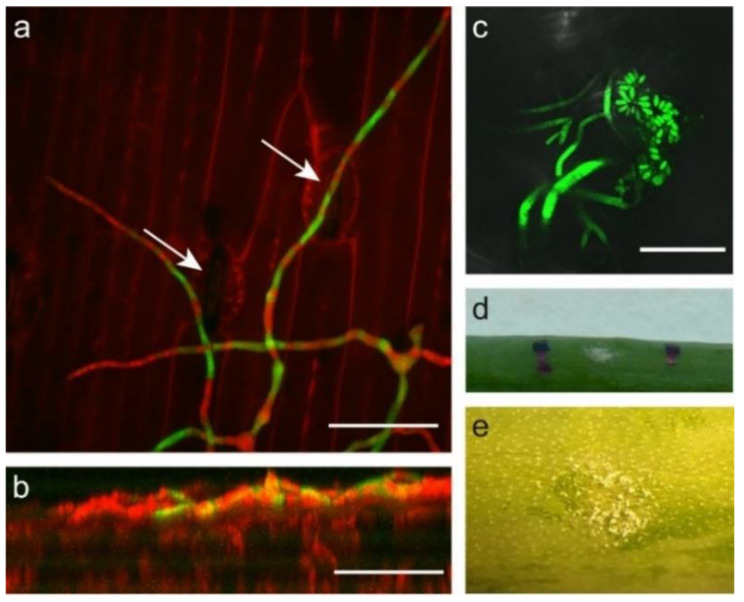
*B. aclada* on onion leaves. (**a**) Top view and (**b**) front view of a 3D projection of confocal laser scanning microscopy images of hyphae growing over and along stomata of an onion leaf. Scale bars represent 50 µm, arrows indicate stomata, *B. aclada*—GFP in green and PI-stained onion cell walls in red. Conidiophores with newly formed spores of *B. aclada* on the surface of an onion leaf observed by (**c**) fluorescence microscopy (scale bar 200 µm), (**d**) on a macroscopic scale, and (**e**) using binocular microscope.

**Figure 4 ijms-22-01865-f004:**
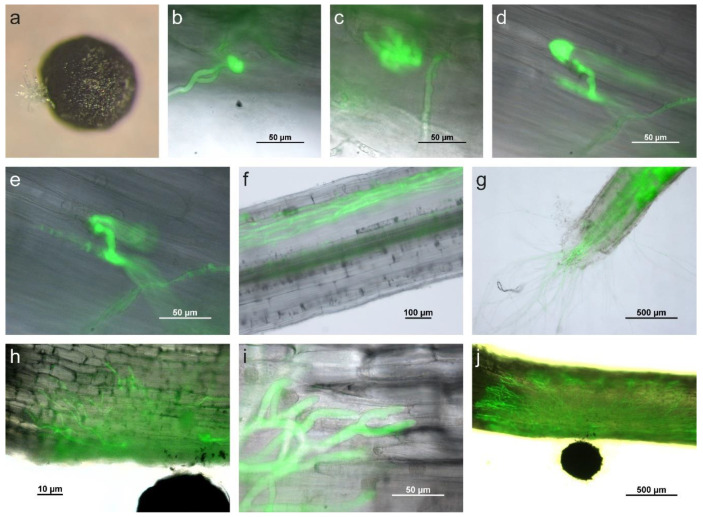
*S. cepivorum* on onion. (**a**) Binocular microscope image of a germinating sclerotium with outgrowth of hyphae. *S. cepivorum* attaching to host surface by forming (**b**) an appressorium-like structure and (**c**) an infection cushion observed by fluorescence microscopy. (**d**,**e**) Invasive hyphae growing inside the onion root 10 h after the formation of an appressorium-like structure. (**f**) Hyphae growing through cortical root tissue and (**g**) eventually growing through the root tip. (**h**) Spreading infection through an onion leaf, with (**i**) disintegrating onion tissue around leading hyphae. (**j**) Hyphae growing over the full width of an onion leaf 36 HPI.

**Figure 5 ijms-22-01865-f005:**
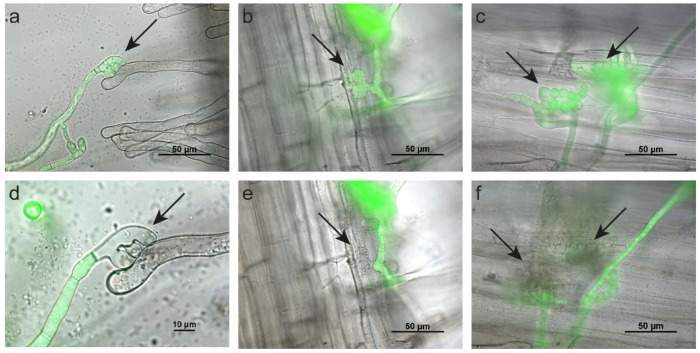
Unsuccessful infection of tomato roots by *S. cepivorum*. Fluorescent microscopical images of (**a**) branching hyphae attached to a tomato-root hair, (**b**) an appressorium-like structure, and (**c**) an infection cushion attached to the tomato-root epidermis. (**d**) 12 h, (**e**) 10 h, and (**f**) 24 h later, infection structures lost their fluorescence, indicated by arrows.

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
