# Peer review of "Visualization of Three Sclerotiniaceae Species Pathogenic on Onion Reveals Distinct Biology and Infection Strategies"

_ijms, 2021, doi:10.3390/ijms22041865_

Round 1
Reviewer 1 Report
In this paper, the authors report “Visualization of three Sclerotiniaceae species pathogenic on onion reveals distinct biology and infection strategies”. The manuscript clarifies the biology of infection of three species of fungi of the family Sclerotiniaceae that are pathogenic in onion, for this they carry out GFP-expressing transformants of all three fungi to visualize the phases of infection These fungi cause very different diseases that pose serious threats to the onion crop; hence the importance of the study and that it is considered of interest to many readers of "IJMS". All technical aspects of the study appear to be performed at an acceptable level. In my view, the manuscript can be accepted for publication, although the authors should review the bibliography, which does not follow the instructions of the journal (format, italics, bold ...).
Author Response
The suggestion of reviewer #1 to review the bibliography style was followed and references were revised according to journal instructions.
Reviewer 2 Report
Dear authors,
I carefully revised the manuscript “Visualization of three Sclerotiniaceae species pathogenic on onion reveals distinct biology and infection strategies”. The manuscript focuses on unveiling the different infection strategies of three pathogenic fungi in onion. The infection process was evaluated by GFP-transformation of the pathogenic fungus and visualization of the pre- and post-infection process by epifluorescence and confocal microscopy. The results show that the three pathogenic fungi have different infection strategies, which are organ-dependent. However, more studies show be conducted in the near future to understand the infection strategies of B. acalda better.
The manuscript is well structured, and the results are well illustrated. However, some pictures (please see comments below) are difficult to see in detail, maybe because of photo quality. If possible, authors should improve this. This is true for supplemental fig 2, Fig 2g and I. Also, the manuscript is easy to read and is well written.
Please consider the following suggestions and considerations for manuscript improving:
Introduction:
#Line 34 – Consider adding the necrotrophic lifestyle in the fungus description, example “The most notorious disease is Botrytis leaf blight, caused by the necrotrophic B. squamosa.” This information will be easily related to lines 42 and 43.
#Line 48- the authors say that “Neck rot is caused by a complex of the three species Botrytis aclada, B. byssoidea and a hybrid species of the former two, called B. allii”. However, in line 31 is suggest B. aclada as a causal agent. Could the authors clarify this? Maybe that B. aclada is the predominant causal agent?
#line 56- concerning S. cepivorum, adding the lifestyle will help related the obtained results and discussion. (similar to what was suggested in #line 34).
#line 65- rephrase “Although all three diseases pose a serious threat to onion, infection mechanisms of especially neck rot and white rot are poorly understood.” To “Although all three diseases pose a serious threat the onion crop, strategies of infection by B. aclada, B. squamosa and S. cepivorum are still poorly understood.”
#line 72 – In my opinion, one of the purposes of studying a plant disease and understanding the infection process/strategies is to establish new disease management and control. Consider including this goal among your owns at the end of the “introduction” section.
Results:
#Section 2.1 – A description of the methodology is included, but no results are presented. The GFP transformation was successfully done, and no abnormalities were observed, but these results are shown in the caption of figure 1. Please include a description of the results obtained here. Maybe a change in the sub-section title could reflect these results
#line 94 – 97 – please rephrase “spores of B. squamosa applied to the surface of onion leaves germinated within the first two hours after inoculation (Figure 2a). The average width and length of B. squamosa conidia measured 15.6μm and 22.6μm respectively, which is in accordance with previously published measurements [21]. “ to “The average width and length of B. squamosa conidia measured 15.6μm and 22.6μm respectively, which is in accordance with previously published measurements [21]. Spores of B. squamosa applied to the surface of onion leaves germinated within the first two hours after inoculation (Figure 2a).”
#line 107 – replace “growing through stomata” by “entering through stomata”
#line 109 – replace “anti-clinal” by “anticlinal” in this line and along with the manuscript.
#line 109-110 – please delete the phrase” Directed growth of germ tubes towards stomata or anti-clinal walls was not observed. “. This is very common since the germs tubes sometimes go around and over the stomata and don’t enter.
#section 2.3 – The authors suggest in the discussion section that B. aclada may enter the host tissue through wounds, since it seems that it does not penetrate through stomata or penetration of cell walls. This is a very interesting hypothesis that should be tested and validated in this section. I suggest including this test since it will contribute for a full explanation of your hypothesis.
#line 153 – about “could be regrown”. What do the authors mean by “regrown”? Do you mean “isolation”?
# line 191- About “throughout the plant”. Does it mean in all directions? If so, this is repeated in the next phrase. Please summarize.
Material and methods
#section 4.3 Plant material and inoculation – There is not a single reference about the protocols used. Is this the first time that these inoculation methods were used?. Please include a reference if it is the case.
#line 348 – please include a paragraph at the end to the sentence “(…) night temperature.”
#line 393-394 – please rephrase. “All statistical analysis was performed in GraphPad 9.0 (Prism). With a two-tailed P value ≤0.05 differences were considered to be statistically significant. Growth rates of WT and GFP-transformants were pairwise compared for B. squamosa, B. aclada and S. cepivorum using unpaired t tests. Germ tube length, curvature and fluorescence intensity at different concentrations PDB in the inoculum were side-by-side analyzed using unpaired t tests. “ to “All statistical analysis was performed in GraphPad 9.0 (Prism). WT and GFP-transformants' growth rates were pairwise compared for B. squamosa, B. aclada and S. cepivorum using unpaired t-tests. Germ tube length, curvature and fluorescence intensity at different concentrations PDB in the inoculum were side-by-side analyzed using unpaired t-tests. With a two-tailed P value ≤0.05 differences were considered to be statistically significant.”
Captions:
#Line 78, Figure 1- please rephrase “ but only GFP-expressing “. The sentence does not make sense.
#line 211, figure 5 – please rephrase “Formation of unsuccessful infection structures of S. cepivorum on tomato roots.”, by ” Unsuccessful infection of S. cepivorum on tomato roots”
Author Response
The suggestions that reviewer #2 made for textual editing are much appreciated, and we have followed these suggestions in the vast majority of cases (see comments below).
Line 34 - ‘the necrotroph’ was inserted instead of ‘the necrotrophic’
Line 48 – Indeed all three Botrytis species (B. aclada, B. allii and B. byssoidea) are able to cause neck rot. This information is inserted in line 31. In the field, predominantly B. aclada and B. allii are identified, but since B. allii is a hybrid species with a genome of 32 instead of 16 chromosomes, we worked with B. aclada.
Line 56 – We did not follow the suggestion to insert the word ‘necrotrophic’ as we think it is inappropriate for this fungus that rots the root system and the bulb surface. We prefer to distinguish the typical cell death symptoms that one can observe on leaf or stem tissues from root rotting processes in the soil.
Line 65 – Sentence was rephrased to ‘Although all three diseases pose a serious threat to onion cultivation, infection mechanisms of especially neck rot and white rot are still poorly understood’.
Line 72 – We agree that developing control strategies is one of the purposes of studying infection biology and changed the sentence to ‘By comparing infection strategies of the three related fungi, we aimed to determine how they cause such distinct diseases on the same host and aimed to provide insights for the development of new control strategies.’.
Section 2.1 – Description of obtained results is included as suggested.
Line 94-97 – Order of sentences is changed as suggested.
Line 107 – Suggestion to replace ‘growing’ by ‘entering’ followed.
Line 109 – ‘anti-clinal’ changed to ‘anticlinal’ in the whole manuscript.
Line 109-110 – Suggestion to delete sentence is adopted.
Section 2.3 –Our hypothesis that B. aclada enters the host tissue through wounds specifically applies to infections under field conditions. When plants are ready for harvest, the bulbs are lifted from the soil and foliage is mechanically cut off. The bulbs are then left to dry on top of the soil for several days before being harvested for cleaning and storage. Performing an inoculation assay with a transgenic fungus on onion bulbs in a simulation of a field situation is illegal. Mimicking such a situation in a lab environment is challenging. The physiological state of the onion plant at the moment of harvesting, as well as the airborne inoculum are impossible to simulate under laboratory conditions.
Line 153 – ‘regrown’ is replaced by ‘isolated’. Indeed the fungus was isolated from leaf segments, but because we very likely observed germination of newly produced conidia instead of growth of mycelium present in leaf tissue, we prefer the term ‘regrowth’ instead of ‘outgrowth’ as discussed in line 155.
Line 191 – ‘throughout the plant’ refers to growth through plant tissue. ‘towards all directions’ refers to radial colonization at a cellular level.
Section 4.3 – Inoculation methods for the purpose of visualization of infection were indeed developed by ourselves and are different from previously published methods for spray inoculations used for e.g. resistance screening.
Line 348 – Paragraph is inserted.
Line 393-394 Section is rephrased as suggested, with a change of word order in the last sentence.
Line 78 – Line is rephrased to ‘and only transformant strains are able to fluoresce’
Line 211 – Sentence is rephrased to ‘Unsuccessful infection of tomato roots by S. cepivorum’.
Round 2
Reviewer 2 Report
Dear authors,
I carefully revised the new version of the manuscript “Visualization of three Sclerotiniaceae species pathogenic on onion reveals distinct biology and infection strategies”. All my questions or suggestions were addressed, and the majority of them were included in this new version. Nevertheless, I only suggest three minor grammatical corrections.
#Line 65 – Concerning the sentence “Although all three diseases pose a serious threat to onion cultivation, infection mechanisms of especially neck rot and white rot are still poorly understood. “. I really suggest rephrasing it to “Although all three diseases pose a serious threat the onion crop, strategies of infection by B. aclada, B. squamosa and S. cepivorum are still poorly understood.”. The reason for this is that the fungus has an infection strategy and disease is the consequence of successful colonization.
#Lines and 80 128 – Caption of figure 1 and 2 – Before p value, please include the reference to the T-student test. The statistical test should be also included in caption of Supplemental figure 1
#Lines 166 and 186 – Please substitute “binocular” by “ binocular microscope”
Best regards
Author Response
" #Line 65 – Concerning the sentence “Although all three diseases pose a serious threat to onion cultivation, infection mechanisms of especially neck rot and white rot are still poorly understood. “.
Authors understand the reasoning to state species names instead of disease names since ‘infection strategy’ relates to the fungus and the disease is the consequence of the infection. We nevertheless prefer not to mention species names because for neck rot we would then need to include all three species names (B. aclada but also B. allii and B. byssoidae), making the sentence unnecessarily difficult to understand. Furthermore, we intend to emphasize the lack of knowledge on neck rot and white rot, whereas leaf blight is relatively well studied. We have now rephrased the sentence to "Although all three diseases pose a serious threat to onion cultivation, the infection biology of these fungi, especially the ones causing neck rot and white rot are still poorly understood"
#Lines and 80 128 – Caption of figure 1 and 2 – Before p value, please include the reference to the T-student test. The statistical test should be also included in caption of Supplemental figure 1.
Information was added as requested.
#Lines 166 and 186 – Please substitute “binocular” by “ binocular microscope”. Substitution made as requested